First record of multi-species synchronous coral spawning from Malaysia

Chelliah Alvin 1
Amar Halimi Bin 1
Hyde Julian 1
Yewdall Katie 2
Steinberg Peter D. 3 4 5
Guest James R. 3 4 jrguest@gmail.com
1 Reef Check Malaysia , Jalan Ampang, Kuala Lumpur , Malaysia
2 Tioman Dive Centre , Kampong Tekek, Tioman Island, Pahang , Malaysia
3 Centre for Marine Bio-Innovation, School of Biological, Earth and Environmental Sciences, University of New South Wales , Sydney, NSW , Australia
4 Advanced Environmental Biotechnology Centre, Nanyang Environment and Water Research Institute, Nanyang Technological University , Singapore , Singapore
5 Sydney Institute of Marine Science , Mosman, NSW , Australia
Thompson Fabiano
Electronic publication date: 2015 Feb 17
Publication date: 2015
Volume: 3
Electronic Location ID: e777
Received 2014 Dec 1; Accepted 2015 Jan 29
Copyright: © 2015 Chelliah et al.
Copyright year: 2015
Copyright holder: Chelliah et al.
License: This is an open access article distributed under the terms of the Creative Commons Attribution License, which permits unrestricted use, distribution, reproduction and adaptation in any medium and for any purpose provided that it is properly attributed. For attribution, the original author(s), title, publication source (PeerJ) and either DOI or URL of the article must be cited.
License URL: https://creativecommons.org/licenses/by/4.0/

Keywords: Multi-species spawning, Coral reef, Pulau Tioman

Funding: Reefcheck Malaysia Research Centre Funding Scheme (RCFS) COY-15-EWI-RCFS/N190-2 This work was funded in part by Reefcheck Malaysia and from the Nanyang Technological University project: “Development of the Advanced Environmental Biotechnology Centre (AEBC)” under the Research Centre Funding Scheme (RCFS), project No. COY-15-EWI-RCFS/N190-2. The funders had no role in study design, data collection and analysis, decision to publish, or preparation of the manuscript.

==============================
Knowledge about the timing and synchrony of coral spawning has important implications for both the ecology and management of coral reef ecosystems. Data on the timing of spawning and extent of synchrony, however, are still lacking for many coral reefs, particularly from equatorial regions and from locations within the coral triangle. Here we present the first documentation of a multi-species coral spawning event from reefs around Pulau Tioman, Peninsular Malaysia, a popular diving and tourist destination located on the edge of the coral triangle. At least 8 coral species from 3 genera (Acropora, Montipora and Porites) participated in multi-species spawning over five nights in April 2014, between two nights before and two nights after the full moon. In addition, two Acropora species were witnessed spawning one night prior to the full moon in October 2014. While two of the Acropora species that reproduced in April (A. millepora and A. nasuta) exhibited highly synchronous spawning (100% of sampled colonies), two other common species (A. hyacinthus and A. digitifera) did not contain visible eggs in the majority of colonies sampled (i.e., <15% of colonies) in either April or October, suggesting that these species spawn at other times of the year. To the best of our knowledge, this is the first detailed documented observation of multi-species coral spawning from reefs in Malaysia. These data provide further support for the contention that this phenomenon is a feature of all speciose coral assemblages, including equatorial reefs. More research is needed, however, to determine the seasonal cycles and extent of spawning synchrony on these reefs and elsewhere in Malaysia.

Introduction

Knowledge about the timing and synchrony of coral spawning has important implications for both the ecology and management of coral reef ecosystems (Guest, 2008). The majority of scleractinian corals (i.e., >60% of species) are hermaphrodites that broadcast gametes for external fertilization (Baird, Guest & Willis, 2009). Broadcast spawning usually occurs annually and can be highly synchronised within populations (Harrison & Wallace, 1990). In addition, within diverse coral assemblages there is often considerable overlap in spawning times among species, leading to extensive multi-species spawning events involving numerous taxa (Babcock et al., 1986). For years it was thought that these remarkable reproductive events were restricted to geographical regions that experience large annual fluctuations in temperature and irradiance (Oliver et al., 1988). More recent research from a wide range of locations, however, has revealed that multi-species coral spawning is likely to be a feature of all speciose coral assemblages (Guest et al., 2005a; Baird, Guest & Willis, 2009; Bouwmeester et al., 2014). A plausible explanation for ubiquitous multi-species spawning can be summarized as follows: Spawning synchrony within broadcast spawning species, driven by external environmental timing cues, is likely to be highly adaptive as it increases the chances of cross fertilization (Babcock et al., 1986; Oliver & Babcock, 1992). Sympatric coral species are likely to respond independently but in a similar manner to the available timing cues—which will lead to overlap in spawning times among species in speciose assemblages (Oliver et al., 1988). Considering that seasonal timing cues (e.g., sea temperature and irradiance) are features of all coastal locations (even at the equator) (Guest et al., 2005a), multi-species spawning is also likely to be a feature of all speciose coral assemblages.

Despite recent advances in knowledge from several previously understudied locations (e.g., Vicentuan et al., 2008; Permata et al., 2012), data on spawning timing and extent of synchrony are still lacking for many coral reefs, particularly from locations within the coral triangle, an area of exceptionally high species diversity encompassing Malaysia, Indonesia, the Philippines and New Guinea (Hoeksema, 2007). The east coast of Peninsular Malaysia has almost 400 recorded scleractinian species, around 70% of species recorded from the entire coral triangle (Huang et al., 2014), and is therefore an area of considerable global importance in terms of marine biodiversity (Harborne et al., 2000). Here we present the first documentation of a multi-species coral spawning event from reefs around Pulau Tioman, Peninsular Malaysia (2°49′09.39″N, 104° 09′ 34.26″E), a popular diving and tourist destination located on the edge of the coral triangle.

Materials and Methods

Evidence from reefs within the coral triangle suggest two coral spawning peaks in March/April and October/November, typically with a minor and a major spawning season for each location (Baird, Guest & Willis, 2009). The spawning times of coral species at sites around Pulau Tioman were examined using three different methods. Firstly, corals were sampled at two fringing reef sites on the west coast of Tioman (TDC House Reef: 2°48′56.47″N 104°09′05.66″E and; Tumuk: 2°47′32.80″N 104°07′22.02″E) on April 12 2014 (3 days before the full moon) and on October 7 2014 (1 day before full moon) to establish the extent of population synchrony within selected coral populations of Acropora. Sampling was done by removing up to three branches from randomly selected, independent (i.e., >5 m apart), replicate colonies of Acropora millepora, A. nasuta, A. hyacinthus and A. digitifera (Table 1) (following Baird, Marshall & Wolstenholme, 2002). Only Acropora colonies >30 cm diameter were sampled to ensure that all were of sufficient size and age to be reproductively mature (see Baria et al., 2012; Wallace, 1985). A. millepora, and A. nasuta were only sampled in April whereas A. hyacinthus, A. digitifera were sampled in April and October. For each colony, the presence or absence of visible pigmented or white eggs was noted in situ by a snorkeler. The presence of pigmented oocytes is indicative of spawning on or close to the date of the next full moon, whereas the presence of visible white eggs indicates that colony will spawn within the next two to three months. Empty colonies have either recently spawned or will not spawn for at least three months (Baird, Marshall & Wolstenholme, 2002). Secondly, to establish the night and time of spawning and the extent of spawning synchrony, we placed small egg-sperm bundle traps made from the bases of upturned plastic water bottles (Guest et al., 2010) over 12 gravid colonies of A. millepora and eight of A. nasuta on 12 April 2014 at TDC House Reef. Gamete traps were also placed over 2 colonies of A. digitifera and, in addition, 2 colonies of A. tenuis that were found to contain pigmented eggs on 7 October 2014. Traps were checked each morning for the presence or absence of released gametes until all colonies had spawned. Finally, in situ observations were made at TDC House Reef by snorkelers on the nights of 13 to 17 April 2014 and on 8 and 9 October 2014 between the hours of 1900 and 2300 to document spawning (approx. 28 h of in situ observation time in total). While the main aim of the direct observations was to establish the timing of spawning of the tagged Acropora colonies, a note was also made of any other coral species spawning during the observation period to assess the extent multi-species spawning at this site.

Table 1 Proportion of colonies sampled containing visible eggs.

Proportion of population with pigmented eggs, white eggs and/or empty colonies in April and October 2014.

Species	Date	Pigmented (%)	White (%)	Empty (%)	n	
Acropora millepora	12/04/2014	100	0	0	26	
Acropora nasuta	12/04/2014	100	0	0	17	
Acropora digitifera	12/04/2014	0	0	100	20	
	7/10/2014	14	0	0	15	
Acropora hyacinthus	12/04/2014	5	0	0	20	
	7/10/2014	0	0	100	15	

Results and Discussion

All sampled colonies of A. millepora and A. nasuta contained visible pigmented eggs when sampled on 12 April 2014 (Table 1). In contrast all sampled colonies of A. digitifera were empty of eggs in April and only 5% of A. hyacinthus colonies contained pigmented eggs in April, with the remainder of the sampled colonies being empty (Table 1). In October, all A. hyacinthus colonies were empty whereas 14% of A. digitifera colonies contained pigmented eggs (Table 1). Examination of the gamete traps showed that 2 colonies (17%) of A. millepora colonies spawned on 13 April, while the remaining tagged colonies of both A. millepora and A. nasuta spawned on 14 April (one night before the full moon) (Fig. 1 and Table 2). Similarly, on October 8, egg-sperm bundles were found in gamete traps placed over two tagged colonies each of A. tenuis and A. digitifera, indicating spawning on October 7 (one night before full moon) for these species. Coral spawning was observed in situ on four of the five nights of observation in April (13, 14, 16 and 17 April) between the hours of 2030 and 2225. No corals were observed spawning on April 15. At least 8 species from 3 genera (Acropora, Montipora and Porites) and 2 families participated in the spawning event (Fig. 1 and Table 2). Night time observations were carried out on October 8 and 9, but no spawning was witnessed directly on these nights. To the best of our knowledge, this is the first detailed documented observation of multi-species coral spawning from reefs in Malaysia. Our data, therefore, support the contention that this phenomenon is a feature of all speciose coral assemblages, including those on equatorial reefs (Baird & Guest, 2009; Guest et al., 2005a). The number of species observed to participate in these events is relatively modest compared to spawning events seen elsewhere (e.g., Babcock et al., 1986). Considering, however, that Pulau Tioman has at least 180 known coral species (Harborne et al., 2000) and that observations were only carried out at one site by two or three observers, it is likely that more extensive surveys will reveal other species participating in these multi-species spawning events.

Figure 1 Multi species coral spawning in Pulau Tioman.

Images of coral spawning in Pulau Tioman showing: (A) a gamete trap containing egg-sperm bundles on Acropora digitifera, (B) spawning of Montipora sp. 1, (C) A. millepora, (D) Porites sp. 2, (E) A. humilis, (F) and a gamete slick on the surface immediately after spawning. (Photos by Alvin Chelliah)

Table 2 Species participation and timing of spawning during multi-species spawning in Pulau Tioman.

Species participation during a multi-species spawning event in April 2014. Spawning nights are relative to date of full moon in 2014 (April 15).

Family	Species	Spawning nights	Spawning time	Gametes released	
Acroporidae	Acropora millepora	−2 to −1	2115 to 2200	B	
	Acropora nasuta	−1	2115 to 2200	B	
	Acropora humilis	−1	2115 to 2200	B	
	Acropora valida	−1	2115 to 2200	B	
	Montipora sp. 1	+ 1 to +2	2030 to 2225	B	
	Montipora sp. 2	+ 2	2030 to 2225	B	
Poritidae	Porites sp. 1	+ 1 to +2	2030 to 2225	S	
	Porites sp. 2	+ 1	2115 to 2225	S	
Notes.

Type of gamete releaseB eggsperm bundles

S sperm

While two species of Acropora (A. millepora and A. nasuta) exhibited highly synchronous spawning in April, two other common species (A. hyacinthus and A. digitifera) did not contain visible eggs in the majority of colonies sampled in either April or October, suggesting spawning at other times of the year for these species. While evidence from nearby locations suggest that March/April and October/November are the two main spawning peaks for this biogeographic region (Baird, Guest & Willis, 2009) extended spawning lasting several months are common on many Indo-Pacific coral reefs (e.g., Bouwmeester et al., 2014). The seasonal timing of spawning (i.e., March/April) for A. millepora and A. nasuta is consistent with observations of spawning seasonality for Acropora species from other locations in Southeast Asia (e.g., Singapore, north-western Philippines, Indonesia) (Guest et al., 2002; Vicentuan et al., 2008; Permata et al., 2012). Seasonal spawning timing within and among coral species is often consistent over broad geographical ranges because individuals are likely to respond in a similar way to environmental timing cues (e.g., sea surface temperature, irradiance) (Willis et al., 1985; Van Woesik, Lacharmoise & Köksal, 2006). The fact, therefore, that A. hyacinthus and A. digitifera did not spawn at this time in Pulau Tioman is surprising as these species have been observed to spawn during the major multi-species spawning period in April in nearby Singapore (Guest et al., 2005a) and Bintan, Indonesia (unpublished data). All sampled colonies were >30 cm in diameter and were found in the same habitat, thus ruling out the possibility that sampled colonies were reproductively immature. It is also interesting to note that the lunar timing of spawning is earlier in Pulau Tioman than for conspecifics in Singapore. For example most species in Singapore spawn between 3 and 6 nights after the full moon (Guest et al., 2002; Guest et al., 2005b) whereas in Pulau Tioman corals spawned between 2 nights before and 2 nights after the full moon.

Clearly, more research is needed to determine the timing and extent of coral spawning synchrony on these reefs and elsewhere in Malaysia. In particular, year-round sampling is required to establish reproductive phenologies for a range of species to determine the cause of timing differences within conspecifics among locations. Comparisons of spawning timing among reefs between the east and west coasts of Peninsular Malaysia are of particular interest, as they experience contrasting monsoon seasons and environmental conditions (Toda et al., 2007).

We are very grateful to the staff at Tioman Dive Centre for field support. All research work was carried out under a memorandum of understanding between Reef Check Malaysia and the Department of Marine Parks Malaysia.

Additional Information and Declarations

Competing Interests

Author Contributions

Field Study Permissions

Alvin Chelliah, Halimi Bin Amar and Julian Hyde are employees of Reef Check Malaysia. Katie Yewdall is an employee of Tioman Dive Centre. Peter D. Steinberg is an employee of the Sydney Institute of Marine Science.

Alvin Chelliah conceived and designed the experiments, performed the experiments, contributed reagents/materials/analysis tools, wrote the paper, prepared figures and/or tables, reviewed drafts of the paper.

Halimi Bin Amar and Katie Yewdall conceived and designed the experiments, performed the experiments, wrote the paper, reviewed drafts of the paper.

Julian Hyde and Peter D. Steinberg conceived and designed the experiments, contributed reagents/materials/analysis tools, wrote the paper, reviewed drafts of the paper.

James R. Guest conceived and designed the experiments, performed the experiments, analyzed the data, contributed reagents/materials/analysis tools, wrote the paper, prepared figures and/or tables, reviewed drafts of the paper.

The following information was supplied relating to field study approvals (i.e., approving body and any reference numbers):

All research work was carried out under a memorandum of understanding between Reef Check Malaysia and the Department of Marine Parks Malaysia.

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
