# Peer review of "First record of multi-species synchronous coral spawning from Malaysia"

_PeerJ, doi:10.7717/peerj.777_

## Round 0.1 · original submission · Minor Revisions

Please provide a revised version of the manuscript and attempt to correct it according to the two referees remarks.

Reviewer 1 ·

Basic reporting

The manuscript “First record of multi-species synchronous coral spawning from Malaysia” by Chelliah et al is an important record about spawning events of eight species of corals from an important touristic location in Malaysia. The manuscript might be accepted to publication but it requires minor revisions.

Authors presented results regarding Montipora and Porites corals in results section but did not mentioned in Material and Methods section. Methods section should be improved (detailed explanation bellow).

The English language in your manuscript must be improved for clarity and readability. I recommend a revision by a native English speaker colleague or by an editing services company.

Abstract:
Authors describe some details about Acropora spawning but don't explicit what species spawned in this event. Please cite all species names in this section.

Introduction:
I suggest including a paragraph or a sentence about spawning events or reproductive strategy of the coral species sampled during this study.

Experimental design

Page 4, lines 52-53: This sentence needs to be improved. If authors decide to write "using a variety of methods", please enumerate the methods.

Page 4, lines 59-60: Why A. millepora and A. nasuta were only sampled in April?

Page 4, lines 65-66: After reading your manuscript I understood that traps were used to determine the number and color of eggs. Please make sure you are describing your methods correctly. I also suggest including a picture of the egg sperm bundle traps in Figure 1 panel.

Validity of the findings

Page 5, line 76: I suggest writing “100%” in full here.

Page 5 lines 77-79: This sentence should be improved.

Page 5 lines 78: There is another way to say that coral colonies had no eggs? Is “Empty” is the best word?

Page 5, line 86-87: Why this date is so important? Please make sure that readers will understand what is your point.

Page 5, lines 87-88: This information must be at Abstract and Material and Methods sections.

Page 5, lines 88-89: In methods sections authors wrote: "Finally, night time observations were made at TDC House Reef by snorkelers on the nights of 13 to 17 April 2014 and on 8 and 9 October 2014 between the hours of 1900 and 2300 to document the timing of spawning and extent of species participation during multi-species spawning.". Please be consistent with dates and species.

Page 5, lines 91-93: Authors inferred that more extensive sampling would reveal more species participating in multi-species spawning events around Pulau Tioman. What make authors infer this? If there is some objective reasons, please discuss it based on literature, if not, please remove this sentence.

Page 6, line 100: Re-write this sentence. It seems bit obvious that other coral species are spawning in other periods of the year. This sentence is really necessary? "Predict" is not the best word in this case. Authors should use speculate instead.

Page 6, lines 103-105: Improve this sentence please.

Additional comments

Present only one table. Join the information of Table 1 and 2 just one.

In table 2 and Figure 1 authors present results of Montipora but in methods section there is no details about the sampling of these colonies.

There is no results in table 1 regarding Montipora or Porites corals. There is some reason for this?

I suggest to include pictures of all studied species in figure 1. Please include a picture of Porites corals’ spawning event.

Please avoid “Table showing…” and “Figure showing…” in tables and figure legends.

·

Basic reporting

The manuscript reports on coral spawning synchronicity for a poorly known region. Text is well written and figures are high quality and very informative. I suggest including additional information about the study sites/sampling design and a more detailed discussion on the hypothesis that multi-species coral spawning is a particular feature of speciose coral assemblages. In my opinion, the manuscript may be accept for publication in PeerJ after a minor review is performed.

Experimental design

- I suggest including a brief justification for the chosen sampling dates (i.e days within the spawning season reported for Acropora corals in the Coral Triangle, cf. Baird et al 2009);
- How many hours were spent for documenting the timing of coral spawning in situ?
- I suggest including additional info in the Material & Methods sections, such as size of sampled colonies and habitat (e.g. reef base versus reef top). Thus, authors could rule out other alternative explanations for the lack of observation of spawning in A. hyacinthus and A. digitifera.

Validity of the findings

Besides the fact that sampling was performed in a poorly known region, the main finding of the manuscript is that multi-species coral spawning is a particular feature of speciose coral assemblages. However, a more comprehensive discussion on this topic is lacking. For example, how many species are known for the studied region in comparison to other areas in which multi-species coral spawning was recorded? Although some information is given for other Indo-Pacific sites, there is no comparison for other biogeographical regions, such as the Caribbean [for example, see Richmond, R. H., & Hunter, C. L. (1990). Reproduction and recruitment of corals: Comparisons among the Caribbean, the Tropical Pacific, and the Red Sea. Marine Ecology Progress Series, 60(1), 185-203 / Szmant, A. M. (1986). Reproductive ecology of Caribbean reef corals. Coral reefs, 5(1), 43-53].

In addition, instead of Baird & Guest (2008), the appropriate reference in the last paragraph (line 112) is as follows:

Guest, J. R., Baird, A. H., Goh, B. P. L., & Chou, L. M. (2005). Seasonal reproduction in equatorial reef corals. Invertebrate Reproduction & Development, 48(1-3), 207-218.

Additional comments

I am sorry for not sending my review earlier. In my opinion, this is a relevant and well presented manuscript that deserves publication in PeerJ after a minor review.

---

## Round 0.2 · accepted · Accept

Dear authors, Dr. Steinberg. congratulations on the accepted paper.